# Environmental exposure to metal mixtures and linear growth in healthy Ugandan children

Emily C. Moody[ORCID][1]*, Elena Colicino[ORCID][1], Robert O. Wright[1], Ezekiel Mupere[2], Ericka G. Jaramillo[3], Chitra Amarasiriwardena[1], Sarah E. Cusick[4]

**1** Department of Environmental Medicine and Public Health, Icahn School of Medicine at Mount Sinai, New York, New York, United States of America, **2** Department of Paediatrics & Child Health, Mulago Hospital, Kampala, Uganda, **3** Department of Medical Education, Icahn School of Medicine at Mount Sinai, New York, New York, United States of America, **4** Department of Pediatrics, University of Minnesota, Minneapolis, Minnesota, United States of America

* emily.moody@mssm.edu

## Abstract

**Data Availability Statement:** All relevant data are within the manuscript and its Supporting Information files.

### Background

Stunting is an indicator of poor linear growth in children and is an important public health problem in many countries. Both nutritional deficits and toxic exposures can contribute to lower height-for-age Z-score (HAZ) and stunting (HAZ < -2).

### Objectives

In a community-based cross-sectional sample of 97 healthy children ages 6–59 months in Kampala, Uganda, we examined whether exposure to Pb, As, Cd, Se, or Zn were associated with HAZ individually or as a mixture.

### Methods

Blood samples were analyzed for a mixture of metals, which represent both toxins and essential nutrients. The association between HAZ and metal exposure was tested using multivariable linear regression and Weighted Quantile Sum (WQS) regression, which uses mixtures of correlated exposures as a predictor.

### Results

There were 22 stunted children in the sample, mean HAZ was -0.74 (SD = 1.84). Linear regression showed that Pb (β = -0.80, p = 0.021) and Se (β = 1.92, p = 0.005) were significantly associated with HAZ. The WQS models separated toxic elements with a presumed negative effect on HAZ (Pb, As, Cd) from essential nutrients with presumed positive effect on HAZ (Se and Zn). The toxic mixture was significantly associated with lower HAZ (β = -0.47, p = 0.03), with 62% of the effect from Pb. The nutrient WQS index did not reach statistical significance (β = -0.47, p = 0.16).

**Funding:** This work was made possible through NIEHS grants T32HD049311 (ECM) and P30ES023515 (ROW), a University of Minnesota School of Medicine Innovation Award (SEC and ECM) and a Doris Duke International Clinical Research Fellowship scholarship (EGJ). The funders had no role in study design, data collection and analysis, decision to publish, or preparation of the manuscript.

**Competing interests:** The authors have declared that no competing interests exist.

## Discussion

Higher blood lead and lower blood selenium level were both associated with lower HAZ. The significant associations by linear regression were reinforced by the WQS models, although not all associations reached statistical significance. These findings suggest that healthy children in this neighborhood of Kampala, Uganda, who have a high burden of toxic exposures, may experience detrimental health effects associated with these exposures in an environment where exposure sources are not well characterized.

## Introduction

Stunting, poor linear growth for age, is a major problem that affects children worldwide particularly in low- and middle-income countries. It has been associated with poor neurodevelopmental and health outcomes in childhood and into adulthood. It has been associated with behavioral problems, cognitive deficits, and greater risk of hypertension and cardiovascular disease later in life [1–5]. The most common causes of stunting include malnutrition, micronutrient deficiencies and infection [5, 6]; however there is growing recognition of the contribution of toxic environmental exposures including lead [7–9]. The negative effects of lead exposure on growth and development are of particular concern because of the universal exposure to children around the globe and disproportionately high exposures in low- and middle-income countries [10, 11]. In countries without strong regulations on industrial use and disposal of toxic substances, the health effects of exposure to growing children are likely increased compared to those in more highly regulated countries. However the actual levels of exposure and their health effects are often unknown due to lack of health surveillance programs and research. The paradigm for understanding exposure sources of toxic metals, metalloids, and metallic elements and the research on cognitive and health effects of these exposures occurs in wealthier countries; yet due to vast differences in housing conditions, urban development and zoning laws, drainage, sanitation, and home-based industry [12], there are likely very different risk factors for exposure and more dangerous exposure patterns facing children in poorer countries. Little is known about toxic metal exposures to young children in Uganda [13, 14] or their effect on stunting in low-and middle-income countries in general [7].

There is rapidly growing interest in the health effects of exposure to mixtures of environmental metals as opposed to those of single exposures because mixed exposure is a better representation of reality. Furthermore, there is evidence that co-exposures to mixtures of chemicals can have synergistically more harmful health effects than individual exposures, especially in the realm of neurodevelopmental effects [15–18]. For health outcomes including stunting and linear growth, we anticipate that in addition to detrimental effects of single exposures, there may be interactions between toxic metals such as lead and essential nutrients such as zinc and selenium [8]. Statistical techniques for analysis of chemical mixtures have been developed to better be able to understand the interactions between individual components of mixtures, and to gain a more comprehensive picture of the health effects of toxicants in concert. For simplicity, we refer to our mixture including lead (Pb), arsenic (As), cadmium (Cd), selenium (Se), and zinc (Zn), as metals, although Zn is a semi-metallic element and Se is a non-metallic element. To our knowledge, statistical techniques for mixtures have not been previously employed to study the effects of metal mixtures on linear growth of children in low income countries.

To determine the effect of exposure to environmental metal mixtures on height-for-age Z-score (HAZ) in this under-studied population, we undertook this analysis of data collected as

part of a community-based study [13] on metal exposure in Ugandan children living in the Katanga urban settlement of Kampala. This study contributes to the understanding of metal exposures to children in a poor neighborhood of Kampala, Uganda and, to our knowledge, it is the first study to investigate the effects of exposure to metal mixtures on linear growth of young children.

## Material and methods

### Study population

In May 2016 community-based cohort of 100 children age 6 to 59 months was recruited using probability sampling method from the Katanga urban settlement near Mulago Hospital in Kampala, Uganda. The community and recruitment procedures are described in detail elsewhere [13]. In short, all children were from the same low-lying community of makeshift temporary structures with poor drainage and garbage management. Inclusion criteria were age 6–59 months, being a permanent resident of the Katanga settlement, and having a caretaker willing to bring the child to Mulago Hospital that day for a medical examination, blood draw, and environmental questionnaire. Children were excluded from the study if they were found to require urgent medical attention or if their caretaker did not speak English or Luganda. Any child requiring urgent medical attention was transported immediately to Mulago Hospital for care. All eligible children and their caretakers were escorted to Mulago Hospital where written informed consent was obtained from the caretaker, a venous blood sample was collected from each child into a metal-free vacutainer tube. A rapid diagnostic test for malaria was completed for each child and positive results were followed up with a Giemsa blood smear. A physical exam was then completed for each child and environmental questionnaire completed by the child's caretaker. The study protocol was approved by the Institutional Review Board of the University of Minnesota, the Research & Ethics Committee of Makerere University School of Biomedical Sciences, and the Uganda National Council for Science and Technology.

### Laboratory analysis

Upon collection, whole blood samples were refrigerated at 4˚C and shipped to New York, USA for analysis. Whole blood samples (1 ml) were acid digested using concentrated nitric acid (2-ml) and hydrogen peroxide (1 ml) at room temperature for 48 hours prior dilution to 10-ml with deionized water and were analyzed using an Agilent 8800 ICP Triple Quad (ICP-QQQ) (Agilent technologies, Inc., Delaware, USA) in MS/MS mode with appropriate cell gases to eliminate molecular ion interferences. Using a method previously described, samples were analyzed for elements including antimony, arsenic, barium, cadmium, cesium, chromium, cobalt, copper, lead, manganese, nickel, selenium, zinc using external calibration with appropriate internal standards (yttrium, indium, tellurium and lutetium) at the Senator Frank R. Lautenberg Environmental Health Sciences Laboratory at the Icahn School of Medicine, Mount Sinai, NY[19, 20]. The limit of detection limits were: 0.013 ng/ml for As, 0.001 ng/ml for Cd, 0.003 ng/ml for Pb, 0.07 ng/ml for Se and 0.1 ng/ml Zn.

### Linear growth

Height and weight were measured during the physical exam for each child. Height-for-age Z-score (HAZ) was calculated for each child using Epi Info version 3.5.1 from Centers for Disease Control and Prevention, which applies height and weight measurements against the Centers for Disease Control/World Health Organization 1978 growth references [21, 22]. Stunting

was defined as height-for-age Z-score (HAZ) > 2 standard deviations below the reference mean for the same sex and age.

## Statistical analyses

Initial descriptive statistics are reported for the whole study population and groups stratified by stunting status. A chi-square test compared the stunted and non-stunted groups for dichotomous population characteristic variables (sex, mother's education, and whether the child had been admitted to the hospital). The Mann-Whitney U test was used for continuous variables (sex and HAZ). Predictors included five metals and metallic elements: lead (Pb), arsenic (As), cadmium (Cd), selenium (Se), and zinc (Zn). Metal data were log2 transformed because of right-skewness for comparability. We tested for differences in the untransformed metal concentrations in the stunted and non-stunted children using a Wilcoxan rank sum test. Analysis by multivariable linear regression modeled HAZ as a continuous outcome. Metals and metallic elements were selected based upon previously described association with stunting (Pb, Se, Zn) [7–9, 23–26], known developmental toxicity (Pb, As, Cd) [27–29] or nutritional importance for healthy growth and development (Se, Zn) [8, 30, 31]. Two subjects were excluded for missing height measurements, and one subject was excluded for errors in heavy metals measurements, resulting in a final sample size of 97 children.

Multivariable linear regression was done to identify which metals may have a significant association with HAZ score. We identified the correlation structure with Pearson's coefficients and we plotted it using a heatmap. Metals were divided into those with expected negative (Pb, As, Cd) and positive effects (Se, Zn) on growth. Associations between these mixtures and HAZ score were analyzed by Weighted Quantile Sum (WQS) regression, using the gWQS package in R. The WQS method analyzes high-dimensional datasets such as environmental exposure mixtures through a weighted index estimating the mixed effect of all predictor variables on the outcome. We tested the relationship between HAZ and a WQS index estimated from ranking exposure concentrations in quintiles (q = 5) for parameter estimation. The weight of each component of the mixture reflects the contribution of that component to the overall effect. We analyzed the mixture of all 5 metals together and for the mixtures based on the presumed direction of association with stunting. We constrained the effect of the mixture to be either positive or negative based upon preliminary analyses and literature. We assumed a linear relationship between exposure and growth. All presented models used 40% of the dataset for training and 60% of the dataset for validation. We assigned 100 bootstrapping steps in each model. All analyses were conducted in R version 3.5.1. All data is available in supporting information (S1 Data).

## Covariates

Maternal educational level was determined by the child's caretaker who answered all questionnaire questions. It was assessed on a six-level scale that was collapsed into two levels (completed primary school or less vs. completed secondary or more) to preserve statistical power. Caretakers also answered whether the child had ever been hospitalized for an illness (yes/no) as a marker of history of major medical illness in the child. Models were not adjusted for child sex or age because the outcome HAZ is generated through a sex and age-specific algorithm. Questionnaires were administered by study staff in Luganda or English.

## Results

### Population characteristics

The final sample included 97 children, and included slightly more boys (n = 51) than girls (n = 46). More than 1 in 5 children were stunted; mean HAZ was below zero and fairly

**Table 1. Demographics of study population.**

| Study Population Characteristics | All n (%) or mean ± SD | Stunted n (%) or mean ± SD | Not Stunted n (%) or mean ± SD | p-value |
|---|---|---|---|---|
| Observations (n) | 97 | 22 | 75 | |
| Child sex | | | | 0.1544 |
| Female | 46 (47.4) | 7 (31.8) | 39 (52.0) | |
| Male | 51 (52.6) | 15 (68.2) | 36 (48.0) | |
| Stunting | | | | |
| Yes | 22 (22.7) | | | |
| No | 75 (77.3) | | | |
| Mother's education | | | | 0.613 |
| Primary school or less | 62 (65.3) | 13 (59.1) | 51 (68.0) | |
| Secondary school or greater | 33 (34.7) | 9 (40.9) | 24 (32.9) | |
| Child ever admitted to hospital | | | | 1.0 |
| Yes | 19 (19.6) | 4 (18.2) | 15 (20.0) | |
| No | 78 (80.4) | 18 (81.8) | 60 (80.0) | |
| Child's age (months) | 28.0 ± 14.9 | 30.7 ± 15.0 | 27.3 ±14.9 | 0.3156 |
| Child's height-for-age Z-score (HAZ) | -0.74 ± 1.84 | -3.12 ± 1.58 | -0.04 ± 1.23 | <0.0001 |

Study population characteristics for the entire study group and subgroups stratified by stunting status. Reported p-value for Chi-Square test for dichotomous variables (sex, mother's education, and whether the child had ever been admitted to the hospital), and for Mann-Whitney U test for continuous variables (age and HAZ).

normally distributed with outliers at -6.31 and -8.68. After careful review, these outliers were determined to be accurate observations and were included in the sample. The age range was 6.1 months to 59.9 months (Table 1). Questionnaires were answered by the child's primary caretaker, who was the mother for the majority (n = 89, 91.8%) of children, a grandparent for 6.2% (n = 6) of children, and the father for 2.1% (n = 2). There were no statistically significant differences between the stunted and non-stunted groups.

## Exposures

All children in our sample had detectable blood lead levels, and 65% (n = 63) of children had blood lead levels ≥5 ug/dL. Untransformed metal exposures are shown in Table 2; there were no significant exposure differences between children with stunting and with no stunting. Exposure means (S1 Table) and correlations are included in supplemental materials (S1 Fig).

**Table 2. Exposures.**

| | All | Stunted | Not Stunted | |
|---|---|---|---|---|
| Metal | Median (IQR) | Median (IQR) | Median (IQR) | p-value |
| Pb (µg/dL) | 5.78 (4.50–7.70) | 6.32 (5.63–7.69) | 5.64 (4.42–7.61) | 0.145 |
| As (µg/L) | 0.23 (0.15–0.33) | 0.25 (0.17–0.36) | 0.22 (0.14–0.32) | 0.395 |
| Cd (µg/L) | 0.084 (0.038–0.130) | 0.084 (0.040–0.120) | 0.084 (0.037–0.140) | 0.860 |
| Se (µg/dL) | 12.20 (10.69–15.02) | 11.53 (9.65–12.13) | 13.05 (10.9–15.49) | 0.0583 |
| Zn (mg/L) | 3.53 (3.02–4.24) | 3.50 (3.09–4.15) | 3.53 (3.02–4.28) | 0.617 |

Metal exposures for the study population (n = 97), and for the stunted (n = 22), and not stunted (n = 75) subgroups. The reported p-value is for a Wilcoxan rank sum test comparing metals exposures in stunted and not stunted populations. All p-values are not significant.

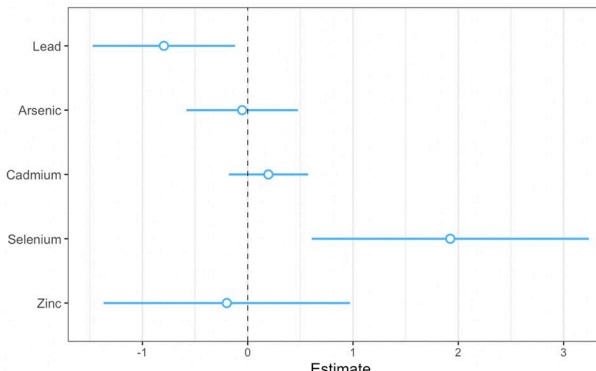

**Fig 1. Effect estimates for individual metals and height-for-age Z-score by multivariable linear regression model.**
A multivariable linear regression model including 5 metals shows a significant negative association between Pb exposure and HAZ score, and a significant positive association between Se exposure and HAZ score. The model was adjusted for level of educational attainment of the child's mother and for a binary variable indicating whether the child had ever been admitted to the hospital.

Full results of metals analysis and the environmental questionnaire are published previously [13].

## Multivariable linear regression

A multivariable model including all five metals and covariates (mother's educational level and whether the child had ever been admitted to the hospital) was used to predict child's HAZ. There was a negative association between Pb and HAZ ($\beta$ = -0.80, p = 0.021), and a positive association between Se and HAZ ($\beta$ = 1.92, p = 0.005) (Fig 1).

## Weighted Quantile Sum regression

Metals were categorized into two groups by direction of the presumed effect on stunting: toxic metals (Pb, As, Cd) and essential nutrients (Se, Zn). The WQS index for toxic metals, assuming a negative association between exposure and HAZ, was statistically significant ($\beta$ = -0.47, p = 0.03), with 62% of the effect attributed to Pb (Fig 2). The essential nutrient WQS index, assuming a positive association between exposure and HAZ, showed that the association was driven by Se, but did not reach statistical significance ($\beta$ = 0.31, p = 0.16) (Fig 3).

A WQS model including all metals and assuming a negative correlation with HAZ did not show a significant association; the relative weights of each component of the mixture showed that the association was driven by Pb and As (S2 Fig). When a positive correlation was assumed using the same mixture, there was a significant association between the WQS index and HAZ (p = 0.027). The relationship was driven primarily by Zn and Se (S3 Fig).

## Discussion

These findings suggest that elevated Pb and lower Se in young children in the Katanga settlement of Kampala, Uganda are associated with decreased HAZ. We demonstrate the novel use of the WQS method to analyze the influence of a mixture of toxic and essential metals and metallic elements as a predictor of HAZ in a low-resource setting, a step in expanding the concept of expososme research to important health outcomes in low resource settings.

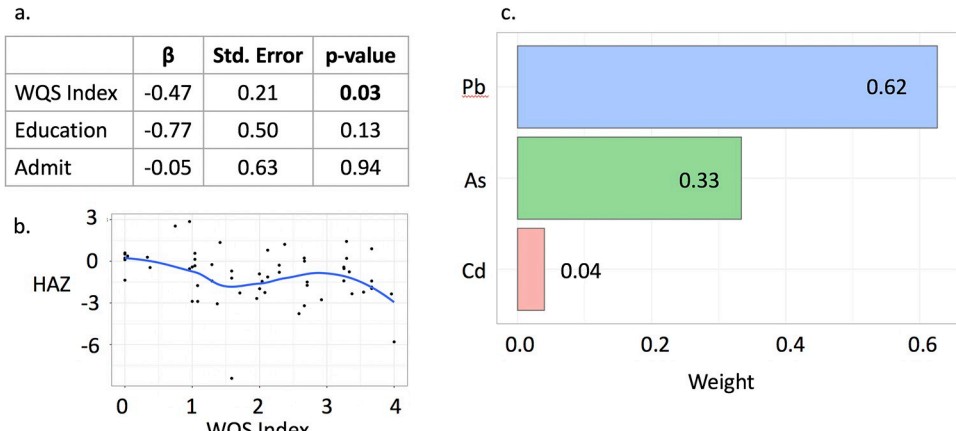

**Fig 2. Weighted Quantile Sum regression for the toxic metal mixture with negative association with HAZ.** a. Results of the regression model for the WQS index of the toxic metals (Pb, As, Cd). b. A locally estimated scatterplot smoothing (LOESS) fit showing the association between the WQS index and HAZ. c. Relative weight of each metal in the mixture.

These findings, the first to examine the effect of metal exposure on linear growth in this population in Uganda, are consistent with previously observed associations between lead exposure and stunting in other populations. In a two-site study of 618 children 20–40 months of age in rural Bangladesh, concurrent blood lead level was associated with increased odds of stunting [7]. The population in this study had a higher prevalence of stunting, at 52.4%, compared to our urban population in Uganda (22.7%) and a lower median blood lead level at 4.2 μg/dL (IQR: 1.7–7.6) vs 5.8 μg/dL (4.5–7.7) in our Ugandan population. Another Bangladeshi study of 729 children under age 2 years in an urban slum environment, of which 39% were stunted and 86.6% had an elevated blood lead level (≥5μg/dL), found no difference in blood lead levels between children who were stunted and those who were not [32]. In addition to childhood concurrent Pb level, prenatal Pb exposures have been associated with decreased

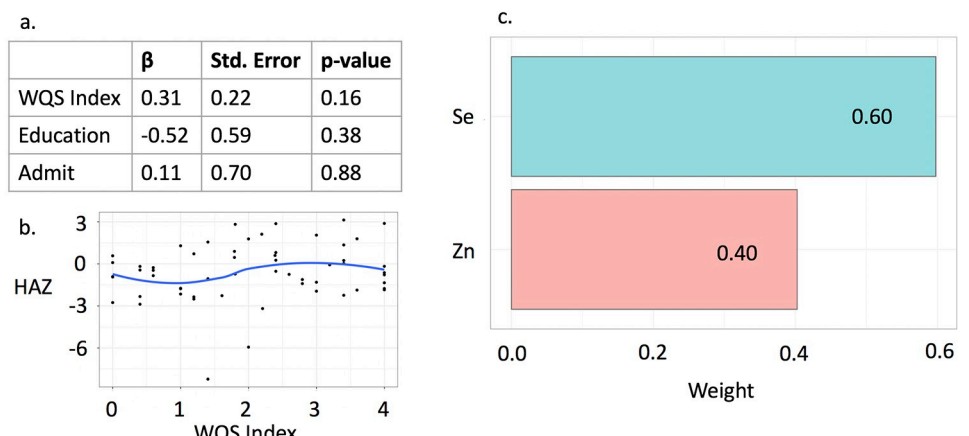

**Fig 3. Weighted Quantile Sum regression for the nutrientmixture with positive association with HAZ.** a. Results of the regression model for the WQS index of the essential nutrients (Se, Zn). b. A locally estimated scatterplot smoothing (LOESS) fit showing the association between the WQS index and HAZ. c. Relative weight of each nutrient in the mixture.

stature. Higher third trimester blood lead levels in mothers was associated with lower HAZ (β-0.10; 95% CI -0.19, -0.01) in a Mexican population of children age 4–6 years [33].

Selenium has a U-shaped toxicity dose-response curve with negative health effects at both deficient and toxic levels. Reported reference ranges vary by population, and prevalence of deficiency and toxicity vary widely based on Se content of the local soil and quality of diet [34, 35]. Our analysis used Se as a continuous variable; we were unable to determine Se deficiency using our measure of Se in whole blood. However, the whole blood Se levels in our population were slightly higher than those reported in 2012 from a population of the same age in Kinshasa, DRC (median 10.7 μg/dL, compared to 12.2 μg/dL in our population), likely due to differences in local Se concentration in soil and drinking water [36]. Low Se is commonly associated with cognitive effects, reduced immune function and other adult diseases related to inflammation; however, it is not commonly associated with childhood growth impairment. Mechanistic plausibility for this association comes from animal studies which have shown that Se deficiency is associated with growth retardation secondary to impaired bone metabolism [23]. There is a need for more research on Se deficiency and its effect on child growth and development [37].

Zn deficiency is more widely recognized to have a negative association with childhood growth and stature [38], although nutrient supplementation trials have had mixed results [39]. In addition to a direct association with stature, recent studies have shown that Zn status may interact with Pb in its effect on linear growth. In a population of 291 Mexican children aged 1–2 years, Zn adequacy was shown to attenuate the negative effect of lead on HAZ [8]. Although it is reasonable to suspect a high prevalence of Zn deficiency in our study population based on local diet and previous studies in similar populations [40–42], whole blood Zn cannot be used to diagnose deficiency so we were unable to determine rates of Zn deficiency in this study. The whole blood Zn levels in our study (median = 3.53 mg/L) were comparable but slightly lower than a population of the same age in Kinshasa, DRC (median = 5.0 mg/L) [36].

Both the negative (Pb) and positive (Se) associations we observed with linear growth in this analysis may have important implications for a child's development far beyond stature. Lower HAZ and stunting (HAZ < -2) have been associated with developmental and health outcomes including cognitive deficits, detachment, and poorer learning as well as lower educational achievement and income [5, 6, 43–45]. Many of the same developmental cognitive effects have been independently associated with higher lead exposures [46] as well as selenium [35] and zinc deficiency [38]; further research is needed to more specifically understand the interactions between exposure to metal mixtures, nutrient status, linear growth, and neurodevelopment.

International child health research and interventions in child development have traditionally focused on poverty, nutritional deficiencies, and quality of learning opportunities [47]. While these factors likely are the strongest determinants of child growth and development in low resource settings, the potential detrimental effects of toxic environmental exposures including metals and air pollution are increasingly recognized as important points for intervention. These factors will continue to grow in importance as climate change progresses and as the number of toxic chemicals in our environment continues to grow. Perhaps the most striking finding in this study is that 65% of children had blood lead levels ≥5 ug/dL, the level at which the US CDC recommends intervention [48]. This research points out the importance of metal exposure in this urban setting and its potential pervasive effects on child development. It is clear that more work to characterize sources of Pb and other metal exposures in this environment is needed. An environmental health questionnaire in this population found no significant association between traditionally recognized risk factors for metal exposure in children (parent occupation, painted walls), and blood metal levels [13]. A previous study from another section of Kampala found increased blood lead level in children with increased proximity to a

local landfill [14]. When risk factors for increased exposure are not understood, it becomes difficult to provide appropriate public health interventions to reduce exposures and reduce risk to children.

## Strengths and limitations

There are multiple limitations of this study to acknowledge. This is an observational community-based sample and the results are not broadly applicable to larger populations of children. We used Epi info version 3.5.1 to derive HAZ scores, which uses growth charts developed by the US CDC/WHO in 1978 based upon a US population of formula-fed infants [21, 49]. Use of this reference for our population of primarily breast-fed babies in Uganda could result in overestimating undernutrition in the babies under 12 months since the rate of growth in formula-fed infants is generally greater than breast-fed infants in the first year of life. We don't expect this significantly affected our results because of the 19 infants under 12 months of age in our sample, there were only 2 with HAZ <-2. The proportion of infants under 12 months with stunting (10.5%) was less than in the over 12-month old group (25.6%) and the overall sample (22.7%). We were also limited by a small sample size, which may have limited our ability to find significant associations; due to this we used HAZ as a continuous rather than dichotomized variable to increase power. We anticipate that advancing this work beyond the small pilot sample will provide opportunities to learn more about interactions between metals, nutrients, and their effect on growth in Ugandan children. If these associations hold true in larger studies, this may represent important areas for intervention in environments known to have increased risk of metals exposures [50]. Finally, our measure of metals in whole blood precludes us from determining Zn or Se deficiency in this population of children.

## Conclusions

This work shows associations between Pb, Se and HAZ, and demonstrates the application of a mixtures analysis methodology to assess the effects of exposures to environmental mixtures in healthy children in urban Kampala Uganda. To date, there is sparse literature on the extent of toxic metal exposure to children in the low- and middle-income countries, the most important sources of exposure in different settings (urban vs rural), and on the expected health effects in these settings. This work suggests opposing effects of toxic and essential metals on HAZ in young children. In order to design more effective interventions to improve early childhood growth and cognitive development, important environmental exposures including lead cannot be ignored. It is important to advance the work to identify sources of contamination from lead and other metals in this poor urban environment, and to work for better protection of children from toxic exposures.

## Supporting information

**S1 Fig. Correlations of metal exposures.** Legend: Correlation plots of metals exposures show the highest correlation between Pb and Cd, and between Se and Zn.
(JPG)

**S2 Fig. WQS model including all metals and assuming a negative correlation with HAZ.** a. Results of the regression model for the WQS index of the metals (Pb, As, Cd, Se, Zn). b. A locally estimated scatterplot smoothing (LOESS) fit showing the association between the WQS index and HAZ. c. Relative weight of each metal in the mixture.
(JPG)

**S3 Fig. WQS model including all metals and assuming a positive correlation with HAZ.** a. Results of the regression model for the WQS index of the metals (Pb, As, Cd, Se, Zn). b. A locally estimated scatterplot smoothing (LOESS) fit showing the association between the WQS index and HAZ. c. Relative weight of each metal in the mixture.
(JPG)

**S1 Table. Unadjusted exposure means.** Caption: Unadjusted means of metal exposures for the study population (n = 97), and for the stunted (n = 22), and not stunted (n = 75) groups.
(DOCX)

**S1 Data.**
(XLSX)

## Author Contributions

**Conceptualization:** Emily C. Moody, Sarah E. Cusick.

**Data curation:** Ericka G. Jaramillo, Chitra Amarasiriwardena, Sarah E. Cusick.

**Formal analysis:** Emily C. Moody, Elena Colicino.

**Funding acquisition:** Sarah E. Cusick.

**Investigation:** Sarah E. Cusick.

**Project administration:** Ericka G. Jaramillo.

**Resources:** Robert O. Wright, Ezekiel Mupere, Sarah E. Cusick.

**Supervision:** Elena Colicino, Robert O. Wright, Ezekiel Mupere, Sarah E. Cusick.

**Visualization:** Emily C. Moody.

**Writing – original draft:** Emily C. Moody.

**Writing – review & editing:** Emily C. Moody, Elena Colicino, Robert O. Wright, Sarah E. Cusick.

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
