## [Decision Letter · Decision Letter 0]

11 Dec 2019

PONE-D-19-30717

Exposure to metal mixtures and linear growth in healthy Ugandan children

PLOS ONE

Dear Dr. Moody,

Thank you for submitting your manuscript to PLOS ONE. After careful consideration, we feel that it has merit but does not fully meet PLOS ONE’s publication criteria as it currently stands. Therefore, we invite you to submit a revised version of the manuscript that addresses the points raised during the review process.

Specifically, major concerns are pointed for methodological description flaws present in the manuscript. I would like to stress the points that reviewer #2 has found. More specifically, data analysis in the context of potential confounding factors need to be discussed.

We would appreciate receiving your revised manuscript by Jan 25 2020 11:59PM. To enhance the reproducibility of your results, we recommend that if applicable you deposit your laboratory protocols in protocols.io, where a protocol can be assigned its own identifier (DOI) such that it can be cited independently in the future. For instructions see: http://journals.plos.org/plosone/s/submission-guidelines#loc-laboratory-protocols

We look forward to receiving your revised manuscript.

Kind regards,

Jose M. Moran

Academic Editor

PLOS ONE

Journal Requirements:

**When submitting your revision, we need you to address these additional requirements:**

**Please ensure that your manuscript meets PLOS ONE's style requirements, including those for file naming. The PLOS ONE style templates can be found at http://www.plosone.org/attachments/PLOSOne_formatting_sample_main_body.pdf and http://www.plosone.org/attachments/PLOSOne_formatting_sample_title_authors_affiliations.pdf**We note that you have stated that you will provide repository information for your data at acceptance. Should your manuscript be accepted for publication, we will hold it until you provide the relevant accession numbers or DOIs necessary to access your data. If you wish to make changes to your Data Availability statement, please describe these changes in your cover letter and we will update your Data Availability statement to reflect the information you provide.

Reviewers' comments:

Reviewer's Responses to Questions

**Comments to the Author**

1. Is the manuscript technically sound, and do the data support the conclusions?

Reviewer #1: Yes

Reviewer #2: Partly

2. Has the statistical analysis been performed appropriately and rigorously? 

Reviewer #1: Yes

Reviewer #2: No

3. Have the authors made all data underlying the findings in their manuscript fully available?

Reviewer #1: Yes

Reviewer #2: Yes

4. Is the manuscript presented in an intelligible fashion and written in standard English?

Reviewer #1: Yes

Reviewer #2: Yes

5. Review Comments to the Author

Reviewer #1: To study whether toxic metals and essential nutrients were associated with growth of children (using height-for-age Z-score, HAZ). There were 97 children enrolled, aged from 6 to 59 months, and their whole blood samples were analyzed for arsenic (As), cadmium (Cd), lead (Pb), selenium (Se), and zinc (Zn) by ICP-MS at the Senator Frank R. Lautenberg Environmental Health Sciences Laboratory at the Icahn School of Medicine, Mount Sinai, NY. The result showed that lower Se and higher Pb blood levels were both associated with lower HAZ.

Abstract line 42, “Lower selenium and higher blood level were both associated with lower HAZ.” What were higher blood levels?

Were all the concentrations of metals in blood analyzed whole blood sample? I know Pb was analyzed with whole blood sample, however, for the other elements, especially, Se and Zn, the standard analysis method are serum. Please clarify this point and provide the method of preparing sample (pre-treatment).

Table 2, please provide the mean and SD as well as median and IQR. Moreover, I suggest to divide into “Stunting” and “Non-stunting” groups, then using independent T test or nonparametric method to test the difference between these 2 groups.

This is an interesting manuscript. Thanks for giving me the chance to review…

Reviewer #2: The manuscript associates the levels of Pb, Cd, As, Zn and Se with stunting in Ugandan children aged 6 to 60 months. Particularly, a negative association between the height-for-age score (HAZ) and Pb in blood and a positive association between HAZ and Se in blood. An new approach introducing a weighted quantile sum (WQS) statistics was applied in the attempt to evaluate the impact of multiple metals in blood on the levels of height for age score.

Some weaknesses are present in the manuscript. Particularly:

1-the living environment and the social status of study children were not included in the analysis, so these potential confounding factors were not taken into account in studying the association between HAZ and levels of metals in blood;

2-the score HAZ is built taking into account the different countries or it was built with US data and adapted to Ugandan children? Could you explain something more about HAZ?

3- the levels of Zn in blood reported in the study are much higher than those expected based on reference values for the general population. How can you explain this?

All this considering, I think that additional information should be included in the paper and that results of the study are only suggestive of a role played by Pb exposure on stunting, so that a major caution should be used in drawing conclusion about the detrimental health effect due to heavy metals

Suggested revisions

Methods

Add information about the year of the field study;

Add the list of metals measured in the original study;

Add information about the living environment and the social status of the study children and explain their effect of the association between HAZ and metals in blood.

Add information about the HAZ score. Is this country specific? In the case this was calculated for US children, do the author have confidence about its applicability in Ugandan children?

The authors describe WQS as a statistical tool for high-dimensional dataset; is the dataset used in this study big enough for the use with WQS?

Results

A R2 = 0.088 is very small; so the model multiple regression model is only marginally explaining the HAZ score.

The figures of the WQS regressions are not very clear. What is the meaning of the graph with the curve? Where is the HAZ score on the graph? Moreover, maybe the graphical representation of this WQS model could be limited to Fig 3 and Fig 4.

Discussion

Add information or hypothesis about sources of exposure to Pb and other metals

Compare the levels of metals in blood in study children and reference values in the general population. From this it should be clear that some issue on Zn in blood is present;

Add info about the derivation of HAZ and its limitation when applied in Ugandan children;

Add the limitations of your study;

Consider to smooth the conclusion of the manuscript, to include the weaknesses of the study and the fact that only Pb seems to exert a negative influence on growth.

6. PLOS authors have the option to publish the peer review history of their article (what does this mean?). If published, this will include your full peer review and any attached files.

Reviewer #1: No

Reviewer #2: No

---

## [Author Response · Author response to Decision Letter 0]

25 Jan 2020

January 17, 2020

Dear Editor and Reviewers, 

Thank you for your thoughtful comments and suggestions on this paper. I have revised the paper according to your suggestions and think that the resulting paper is both stronger and clearer. I appreciate your time and contributions to this publication. 

Sincerely, 

Emily Moody

 

Reviewer #1: 

To study whether toxic metals and essential nutrients were associated with growth of children (using height-for-age Z-score, HAZ). There were 97 children enrolled, aged from 6 to 59 months, and their whole blood samples were analyzed for arsenic (As), cadmium (Cd), lead (Pb), selenium (Se), and zinc (Zn) by ICP-MS at the Senator Frank R. Lautenberg Environmental Health Sciences Laboratory at the Icahn School of Medicine, Mount Sinai, NY. The result showed that lower Se and higher Pb blood levels were both associated with lower HAZ.

Abstract line 42, “Lower selenium and higher blood level were both associated with lower HAZ.” What were higher blood levels?

Were all the concentrations of metals in blood analyzed whole blood sample? I know Pb was analyzed with whole blood sample, however, for the other elements, especially, Se and Zn, the standard analysis method are serum. Please clarify this point and provide the method of preparing sample (pre-treatment).

Response: Yes, the analysis method included a single sample preparation from whole blood. Further information on sample preparation has been included, and references from previous publications employing this method. In addition, the inability to comment on Se or Zn deficiency with this sample source was commented on in the limitations section as well as the results sections. To limit the risk to the young children who participated we chose to take a single blood sample to minimize the amount of blood drawn, so we did not have enough to fractionate and analyze whole blood and serum samples.

Updated manuscript text: Page 6, Line 120-131: “Upon collection whole blood samples were refrigerated at 4oC and shipped to New York, USA for analysis. Whole blood samples (1 ml) were acid digested using concentrated nitric acid (2-ml) and hydrogen peroxide (1 ml) at room temperature for 48 hours prior dilution to 10-ml with deionized water and were analyzed using an Agilent 8800 ICP Triple Quad (ICP-QQQ) (Agilent technologies, Inc., Delaware, USA) in MS/MS mode with appropriate cell gases to eliminate molecular ion interferences. Using a method previously described, samples were analyzed for heavy metals including antimony, arsenic, barium, cadmium, cesium, chromium, cobalt, copper, lead, manganese, nickel, selenium, zinc using external calibration with appropriate internal standards (yttrium, indium, tellurium and lutetium) at the Senator Frank R. Lautenberg Environmental Health Sciences Laboratory at the Icahn School of Medicine, Mount Sinai, NY[CHITRA REFS PMID: 30849576, 31357156].”

Page 14, Line 302-303: “we were unable to determine Se deficiency using whole blood Se”

Page 15, Line 316-317: “whole blood Zn cannot be used to diagnose deficiency so we were unable to determine rates of Zn deficiency in this study.”

Page 17, Lines 353-354: “Finally, our measure of metals in whole blood precludes us from determining Zn or Se deficiency in this population of children.”

Table 2, please provide the mean and SD as well as median and IQR. Moreover, I suggest to divide into “Stunting” and “Non-stunting” groups, then using independent T test or nonparametric method to test the difference between these 2 groups.

Response: The mean and SD have been added to the table. Another column for the p-value from a wilcoxan rank sum test was added (stunted children vs non-stunted children) showing no significant differences between the stunted and non-stunted groups for any of the heavy metal concentrations. 

This is an interesting manuscript. Thanks for giving me the chance to review…

 

Reviewer #2: 

The manuscript associates the levels of Pb, Cd, As, Zn and Se with stunting in Ugandan children aged 6 to 60 months. Particularly, a negative association between the height-for-age score (HAZ) and Pb in blood and a positive association between HAZ and Se in blood. A new approach introducing a weighted quantile sum (WQS) statistics was applied in the attempt to evaluate the impact of multiple metals in blood on the levels of height for age score.

Some weaknesses are present in the manuscript. Particularly:

1-the living environment and the social status of study children were not included in the analysis, so these potential confounding factors were not taken into account in studying the association between HAZ and levels of metals in blood;

2-the score HAZ is built taking into account the different countries or it was built with US data and adapted to Ugandan children? Could you explain something more about HAZ?

3- the levels of Zn in blood reported in the study are much higher than those expected based on reference values for the general population. How can you explain this?

All this considering, I think that additional information should be included in the paper and that results of the study are only suggestive of a role played by Pb exposure on stunting, so that a major caution should be used in drawing conclusion about the detrimental health effect due to heavy metals.

Suggested revisions

Methods

Add information about the year of the field study;

Updated manuscript text: Page 6, Line 99 “In May 2016, “

Add the list of metals measured in the original study;

Updated manuscript text: Page 7, Lines 119-121 “…samples were analyzed for antimony, arsenic, barium, cadmium, cesium, chromium, cobalt, copper, lead, manganese, nickel, selenium, zinc were measured by LC-tandem mass spectrometry in the Senator Frank R. Lautenberg Environmental Health Sciences Laboratory at the Icahn School of Medicine, Mount Sinai, NY. 

Add information about the living environment and the social status of the study children and explain their effect of the association between HAZ and metals in blood.

Response: The neighborhood where the children live, and from which they were recruited, is described in detail in the referenced publication. 

Updated Manuscript text: Page 6, Lines 96-98 “The community and recruitment procedures are described in detail elsewhere (1). In short, all children were from the same low-lying community of makeshift temporary structures with poor drainage and garbage management.”

Add information about the HAZ score. Is this country specific? In the case this was calculated for US children, do the author have confidence about its applicability in Ugandan children?

Response: We calculated HAZ scores using Epi info version 3.5.1 These standards were developed form normal formula-fed children in the USA. When applied to this population in Uganda, there may be overestimation of estimates of poor growth in the <1 yo children because breastfed babies grow more slowly than formula fed babies. This has been added to the discussion of limitations of the study. 

Updated manuscript text: Page 16, Lines 310-313 “We used Epi info version 3.5.1 to derive HAZ scores, which is based upon a sample of formula-fed babies in the USA in 1978, which could overestimate undernutrition in our sample since the rate of growth in formula-fed infants is greater than breast-fed in the first year of life.”

The authors describe WQS as a statistical tool for high-dimensional dataset; is the dataset used in this study big enough for the use with WQS?

Response: WQS is a statistical tool that can handle from small to high dimensional datasets. We have previously implemented WQS using a small dataset and found good performance of the WQS under those conditions (2). A WQS extension for high-dimension data is available when needed (3). This study did have a small sample size. To overcome the challenges of the smaller sample size, we use 40% of the observations to train the model and 60% of the observations to validate the model. We take confidence that the model performance is satisfactory from the example of another small dataset (N=200) that had a much higher-dimension dataset with a ratio of 5:1 of variables to observations (4). Our data set had a variable: observation ratio of 19.5:1.

Results

A R2 = 0.088 is very small; so the model multiple regression model is only marginally explaining the HAZ score.

Response: In many environmental studies, the R2 are generally small. We expect that had we been able to account for nutritional factors we would have been able to much better account for HAZ variation in the children. However, this R2 is very small which may be due to other conditions we are not aware of. To reduce confusion, we have removed this sentence from the manuscript. 

The figures of the WQS regressions are not very clear. What is the meaning of the graph with the curve? Where is the HAZ score on the graph? 

Response: Please excuse my oversight that the y-axis label for the graph with the curve was incorrectly labeled. It should say HAZ (our outcome) and this has been changed in all figures. The curve is a LOESS (locally estimated scatterplot smoothing) fit which represents the association between the WQS index and the outcome variable. The outcome variable is adjusted for all covariates, and a summary of each variables’ relative weight within that index.

Moreover, maybe the graphical representation of this WQS model could be limited to Fig 3 and Fig 4.

Response: This is reasonable. The results from the WQS analyses divided by direction of effect are the main findings for this paper, so the WQS figures for those findings remain as part of the main text. The secondary findings including all 5 metals in the WQS analyses have been moved to supporting information.

Discussion

Add information or hypothesis about sources of exposure to Pb and other metals

Response: Description of the environmental questionnaire and discussion of possible sources fo Pb and other metal exposure was in a previous publication (1). However, I have added a small amount of discussion of this issue. 

Updated Manuscript Text: Page 16, Lines 297-303 “It is clear that more work to characterize sources of Pb and other heavy metal exposures in this environment is needed. An environmental health questionnaire in this population found no significant association between traditionally recognized heavy metal exposure risk factors in children (parent occupation, painted walls), and blood metal levels (1). One study from another section of Kampala found increased blood lead level in children with increased proximity to a local landfill (5).”

Compare the levels of metals in blood in study children and reference values in the general population. From this it should be clear that some issue on Zn in blood is present;

Response: It was the purpose of a previous publication to discuss the values in detail(1). Also, because our analysis of all metals was in whole blood, we are unable to comment on either Zn or Se deficiency. However, we do compare our results with a publication that similarly analyzed heavy metals in whole blood of children age 6-59 months from Kinshasa DRC. 

Updated manuscript text: Page 15, Lines 274-279 “Although it is reasonable to suspect a high prevalence of Zn deficiency in our study population based on local diet and previous studies in similar populations (6-8), whole blood Zn cannot be used to diagnose deficiency so we were unable to determine rates of Zn deficiency in this study. The whole blood Zn levels in our study (median = 3.53 mg/L) were comparable but slightly lower than a population of the same age in Kinshasa, DRC (median = 5.0 mg/L) (9). “

Add info about the derivation of HAZ and its limitation when applied in Ugandan children;

Updated manuscript text: Page 16, Lines 311-314 “We used Epi info version 3.5.1 to derive HAZ scores, which is based upon a sample of formula-fed babies in the USA in 1978, which could overestimate undernutrition in our sample since the rate of growth in formula-fed infants is greater than breast-fed in the first year of life.”

Add the limitations of your study; Consider to smooth the conclusion of the manuscript, to include the weaknesses of the study and the fact that only Pb seems to exert a negative influence on growth.

Response: The limitations section was updated and expanded. Page 16

References: 

1. Cusick SE, Jaramillo EG, Moody EC, Ssemata AS, Bitwayi D, Lund TC, et al. Assessment of blood levels of heavy metals including lead and manganese in healthy children living in the Katanga settlement of Kampala, Uganda. BMC Public Health. 2018;18(1):717.

2. Carrico C, Gennings C, Wheeler DC, Factor-Litvak P. Characterization of Weighted Quantile Sum Regression for Highly Correlated Data in a Risk Analysis Setting. J Agric Biol Environ Stat. 2015;20(1):100-20.

3. Curtin P, Kellogg J, Cech N, Gennings C. A random subset implementation of weighted quantile sum (WQSRS) regression for analysis of high-dimensional mixtures. Communications in Statistics - Simulation and Computation. 2019.

4. Team CMA, Mazzella M, Sumner SJ, Gao S, Su L, Diao N, et al. Quantitative methods for metabolomic analyses evaluated in the Children's Health Exposure Analysis Resource (CHEAR). J Expo Sci Environ Epidemiol. 2020;30(1):16-27.

5. Graber LK, Asher D, Anandaraja N, Bopp RF, Merrill K, Cullen MR, et al. Childhood lead exposure after the phaseout of leaded gasoline: an ecological study of school-age children in Kampala, Uganda. Environ Health Perspect. 2010;118(6):884-9.

6. Walker BE, Kelleher J. Plasma whole blood and urine zinc in the assessment of zinc deficiency in the rat. J Nutr. 1978;108(10):1702-7.

7. Ndeezi G, Tumwine JK, Bolann BJ, Ndugwa CM, Tylleskar T. Zinc status in HIV infected Ugandan children aged 1-5 years: a cross sectional baseline survey. BMC Pediatr. 2010;10:68.

8. Bitarakwate E, Mworozi E, Kekitiinwa A. Serum zinc status of children with persistent diarrhoea admitted to the diarrhoea management unit of Mulago Hospital, Uganda. Afr Health Sci. 2003;3(2):54-60.

9. Tuakuila J, Kabamba M, Mata H, Mata G. Toxic and essential elements in children's blood (<6 years) from Kinshasa, DRC (the Democratic Republic of Congo). J Trace Elem Med Biol. 2014;28(1):45-9.

---

## [Decision Letter · Decision Letter 1]

17 Feb 2020

PONE-D-19-30717R1

Exposure to metal mixtures and linear growth in healthy Ugandan children

PLOS ONE

Dear Dr. Moody,

Thank you for submitting your manuscript to PLOS ONE. After careful consideration, we feel that it has merit but does not fully meet PLOS ONE’s publication criteria as it currently stands. Therefore, we invite you to submit a revised version of the manuscript that addresses the points raised during the review process.

The manuscript has improved but there are still points that deserve the rigorous attention of the authors, otherwise the manuscript cannot be published. For this reason, although the authors have already undertaken a review of the manuscript, I believe that the manuscript still requires further revision.

We would appreciate receiving your revised manuscript by Apr 02 2020 11:59PM. To enhance the reproducibility of your results, we recommend that if applicable you deposit your laboratory protocols in protocols.io, where a protocol can be assigned its own identifier (DOI) such that it can be cited independently in the future. For instructions see: http://journals.plos.org/plosone/s/submission-guidelines#loc-laboratory-protocols

We look forward to receiving your revised manuscript.

Kind regards,

Jose M. Moran

Academic Editor

PLOS ONE

Reviewers' comments:

Reviewer's Responses to Questions

**Comments to the Author**

1. If the authors have adequately addressed your comments raised in a previous round of review and you feel that this manuscript is now acceptable for publication, you may indicate that here to bypass the “Comments to the Author” section, enter your conflict of interest statement in the “Confidential to Editor” section, and submit your "Accept" recommendation.

Reviewer #1: (No Response)

Reviewer #2: All comments have been addressed

2. Is the manuscript technically sound, and do the data support the conclusions?

Reviewer #1: Yes

Reviewer #2: Partly

3. Has the statistical analysis been performed appropriately and rigorously? 

Reviewer #1: Yes

Reviewer #2: No

4. Have the authors made all data underlying the findings in their manuscript fully available?

Reviewer #1: Yes

Reviewer #2: No

5. Is the manuscript presented in an intelligible fashion and written in standard English?

Reviewer #1: Yes

Reviewer #2: Yes

6. Review Comments to the Author

Reviewer #1: Table 2 of this version needed still to improve. I saw the authors said adding a column of p-values to show the test (Wilcoxan rank sum test) of stunted children vs non-stunted children. However, I did not read any of this information in the table 2 nor in the text. The readers will not understand what for these p-values were.

I still suggest that sepreate into 2 groups (stunted children vs non-stunted children) of the metal data distributions for the table 2, as I suggested previously.

Reviewer #2: The manuscript has been revised and improved, but still some points rise concern and need interventions.

Particularly:

1-the authors state that they measured heavy metals. Unfortunately this is questionable, as

• As is a semi metallic element,

• Se in a non metallic element

Sorry for not having pointed this before. Please revise the text considering this issue.

Line 42 in abstract. The negative effect is presumed as well.

Discussion: I still think that the results of this study do not demonstrate, but suggest that children are experiencing detrimental effects associated with exposure to environmental pollutants. This comment was previously given, but authors did not take into account it. Moreover, the effect seems to be associated with Pb and not with other pollutants.

Paragraph Linear Growth. Please add here the information about the HAZ score. That is:

1-this was developed for US children fed with formula.

2. this may be not suitable for breast-feed children up to 1 year. Please specify where we can find the algorithm and apply it. Add references too.

Given that the HAZ index in not working very well for children younger than 1 y, could you please state how many of the stunting children were in this range of age?

Table 1: divide the table and compare the groups of stunting and non-stunting children.

Table 2 and t-test between stunted and non stunted children. Please, divide children based on their classification of stunting /non-stunting and compare the groups. IN the present version, a p value is given, but it is not clear why. Specify it in the legend.

The correlation between metals is not relevant to the study; I propose to remove Figure 1, especially considering that the data on metals were published before.

Discussion.

The study does not demonstrate but suggest, a role of lead on stunting in Ugandan children; this limit the generalization to the other investigated metals.

Moreover, it should be clarified how many children, classified as stunting, were below 12 months, considering the limitation of the HAZ index for those children.

In general, author should add some additional info in the manuscript, and use more caution in the interpretation of their results.

7. PLOS authors have the option to publish the peer review history of their article (what does this mean?). If published, this will include your full peer review and any attached files.

Reviewer #1: Yes: Hung-Yi Chuang

Reviewer #2: No

---

## [Author Response · Author response to Decision Letter 1]

28 Feb 2020

Dear Dr. Jose Moran and Reviewers, 

Thank you for the time you have put into reviewing this paper. I appreciate the comments and have done my best to fully and thoughtfully respond to each. I hope that you find the responses have strengthened the manuscript as I certainly do. I look forward to any further communication. 

In the following, please find the numbered comments from the reviewers followed by bulleted responses. 

Sincerely, 

Emily Moody

Reviewer #1: 

1. Table 2 of this version needed still to improve. I saw the authors said adding a column of p-values to show the test (Wilcoxon rank sum test) of stunted children vs non-stunted children. However, I did not read any of this information in the table 2 nor in the text. The readers will not understand what for these p-values were.

• In response to the sum of comments on Table 2, and in an effort to make the table easy to read, I have decided to use the log2 transformed metals measures and to provide the mean (SD) rather than both mean (SD) and median (IQR). The t-test to compare stunted and non-stunted groups was done and is described in the methods section as well as the caption for the table. 

• The original table with unadjusted metals measures, including both mean (SD) and median (IQR), and including stunted and non-stunted groups and the appropriate Wilcoxon rank sum test is now included in the supplemental material. 

• METHODS - LINE 144-146: “Heavy metals data were log2 transformed because of right-skewness for comparability. We tested for differences in the transformed heavy metal concentrations in the stunted and non-stunted children using a t-test.”

• RESULTS – LINE 199-202: “Transformed heavy metal exposures are shown in Table 2; there were no significant differences in heavy metal exposures between children with stunting and with no stunting. Untransformed heavy metal exposures (S1 Table) and correlations between heavy metal exposures are included in supplemental materials (S1Fig).”

• CAPTION – LINE 205-208: “Heavy metal exposures for the study population (n=97), and for the stunted (n=22), and not stunted (n=75) stratified groups. Heavy metals data were log2 transformed because of right-skewness. The reported p-value is for a t-test comparing heavy metals exposures in stunted and not stunted populations. All p-values are not significant.” 

2. I still suggest that sepreate into 2 groups (stunted children vs non-stunted children) of the metal data distributions for the table 2, as I suggested previously. 

• Thank you for the suggestion, I’m sorry I did not clearly understand your previous request. This has been included. 

Reviewer #2: The manuscript has been revised and improved, but still some points rise concern and need interventions. Particularly:

1. the authors state that they measured heavy metals. Unfortunately this is questionable, as

• As is a semi metallic element,

• Se in a non metallic element

Sorry for not having pointed this before. Please revise the text considering this issue.

• Thank you for the comment. I have included a brief definition and citation in the introduction for readers who may be less familiar. 

• LINE 74-76: “Heavy metals are a diverse group of naturally occurring metallic elements with high density and toxicity to humans; some are also essential nutrients and are referred to as essential heavy metals (1).”

• I have updated the title and text to more clearly say “heavy metal” rather than “metal” for greater accuracy. 

2. Line 42 in abstract. The negative effect is presumed as well.

• This is true. Text was updated to read: 

• LINE 36-38 “The WQS models separated nutritional metals with presumed positive effect on HAZ (Se and Zn) from toxic metals with a presumed negative effect on HAZ (Pb, As, Cd).”

3. Discussion: I still think that the results of this study do not demonstrate, but suggest that children are experiencing detrimental effects associated with exposure to environmental pollutants. This comment was previously given, but authors did not take into account it. Moreover, the effect seems to be associated with Pb and not with other pollutants.

• I agree with your observation and interpretation. I have revised in an attempt to change all similar language to more appropriately express that. 

• LINE 44-47 “These findings suggest that healthy children in this neighborhood of Kampala, Uganda, who have a high burden of toxic heavy metal exposure, may experience detrimental health effects associated with these exposures in an environment where exposure sources are not well characterized.”

• LINE 253-254: “These findings suggest that elevated Pb and lower Se in young children in the Katanga region of Kampala, Uganda are associated with decreased HAZ.”

4. Paragraph Linear Growth. Please add here the information about the HAZ score. That is:

1-this was developed for US children fed with formula.

2. this may be not suitable for breast-feed children up to 1 year. Please specify where we can find the algorithm and apply it. Add references too.

• Thank you for attending to this important detail. The following text was changed: 

• LINE 131-134: “Height-for-age Z-score (HAZ) was calculated for each child using Epi Info version 3.5.1 from Centers for Disease Control and Prevention, which applies height and weight measurements against the Centers for Disease Control/World Health Organization 1978 growth references (2, 3).”

• LINE 329-337 “We used Epi info version 3.5.1 to derive HAZ scores, which uses growth charts developed by the US CDC/WHO in 1978 based upon a US population of formula-fed infants (2, 4). Use of this reference for our population of primarily breast fed babies in Uganda could result in overestimating undernutrition in the babies under 12 months since the rate of growth in formula-fed infants is generally greater than breast-fed infants in the first year of life. We don’t expect this significantly affected our results because of the 19 infants under 12 months of age in our sample, there were only 2 with HAZ <-2. The proportion of infants under 12 months with stunting (10.5%) was less than in the over 12-month old group (25.6%) and the overall sample (22.7%).”

5. Given that the HAZ index in not working very well for children younger than 1 y, could you please state how many of the stunting children were in this range of age?

• Of course. There were 19 children younger than 1 year, two (11%) of whom were stunted. Of the 78 children older than one year, 20 (26%) were stunted. 

• LINES 331-337 “We don’t expect this significantly affected our results because of the 19 infants under 12 months of age in our sample, there were only 2 with HAZ <-2. The proportion of infants under 12 months with stunting (10.5%) was less than in the over 12-month old group (25.6%) and the overall sample (22.7%).” 

6. Table 1: divide the table and compare the groups of stunting and non-stunting children.

• The table has been updated. A chi-square test compared the stunted and non-stunted groups for dichotomous population characteristic variables (sex, mother’s education, and whether the child had been admitted to the hospital). The Mann-Whitney U test was used for continuous variables (sex and HAZ). This p-value is reported with clear explanation in the methods section and in the table caption. 

• LINE 139-143: “Initial descriptive statistics were for the whole study population and groups stratified by stunting status. A chi-square test compared the stunted and non-stunted groups for dichotomous population characteristic variables (sex, mother’s education, and whether the child had been admitted to the hospital). The Mann Whitney U test was used for continuous variables (sex and HAZ).”

• CAPTION – LINE 193-196: “Study population characteristics for the entire study group and groups stratified by stunting status. Reported p-value for Chi-Square test for dichotomous variables (sex, mother’s education, and whether the child had ever been admitted to the hospital), and for Mann-Whitney U test for continuous variables (age and HAZ).” 

7. Table 2 and t-test between stunted and non stunted children. Please, divide children based on their classification of stunting /non-stunting and compare the groups. IN the present version, a p value is given, but it is not clear why. Specify it in the legend.

• In response to the sum of comments on Table 2, and in an effort to make the table easy to read, I have decided to use the log2 transformed metals measures and to provide the mean (SD) rather than both mean (SD) and median (IQR). The t-test to compare stunted and non-stunted groups was done and is described in the methods section as well as the caption for the table. 

• The original table with unadjusted metals measures, including both mean (SD) and median (IQR), and including stunted and non-stunted groups and the appropriate Wilcoxon rank sum test is now included in the supplemental material. 

• METHODS - LINE 144-146: “Heavy metals data were log2 transformed because of right-skewness for comparability. We tested for differences in the transformed heavy metal concentrations in the stunted and non-stunted children using a t-test.”

• RESULTS – LINE 199-202: “Transformed heavy metal exposures are shown in Table 2; there were no significant differences in heavy metal exposures between children with stunting and with no stunting. Untransformed heavy metal exposures (S1 Table) and correlations between heavy metal exposures are included in supplemental materials (S1Fig).”

• CAPTION – LINE 205-208: “Heavy metal exposures for the study population (n=97), and for the stunted (n=22), and not stunted (n=75) stratified groups. Heavy metals data were log2 transformed because of right-skewness. The reported p-value is for a t-test comparing heavy metals exposures in stunted and not stunted populations. All p-values are not significant.”

8. The correlation between metals is not relevant to the study; I propose to remove Figure 1, especially considering that the data on metals were published before.

• The correlation plot has been moved to the supplemental materials where a reader may find the information if they are interested. 

9. Discussion.

The study does not demonstrate but suggest, a role of lead on stunting in Ugandan children; this limit the generalization to the other investigated metals.

Moreover, it should be clarified how many children, classified as stunting, were below 12 months, considering the limitation of the HAZ index for those children.

In general, author should add some additional info in the manuscript, and use more caution in the interpretation of their results.

• Thank you for your contribution to this manuscript. These suggestions have been taken into account and answered in the above numerated items. 

References:

1. Tchounwou PB, Yedjou CG, Patlolla AK, Sutton DJ. Heavy metal toxicity and the environment. Exp Suppl. 2012;101:133-64.

2. Center for Disease Control. Use of World Health Organization and CDC Growth Charts for Children Aged 0–59 Months in the United States. September 10, 2010.

3. U.S. Department of Health & Human Services. Epi Info: Center for Surveillance, Epidemiology & Laboratory Services; [Available from: https://www.cdc.gov/epiinfo/index.html.

4. Dibley MJ, Goldsby JB, Staehling NW, Trowbridge FL. Development of normalized curves for the international growth reference: historical and technical considerations. Am J Clin Nutr. 1987;46(5):736-48.

---

## [Decision Letter · Decision Letter 2]

10 Mar 2020

PONE-D-19-30717R2

Exposure to heavy metal mixtures and linear growth in healthy Ugandan children

PLOS ONE

Dear Dr. Moody,

Thank you for submitting your manuscript to PLOS ONE. After careful consideration, we feel that it has merit but does not fully meet PLOS ONE’s publication criteria as it currently stands. Therefore, we invite you to submit a revised version of the manuscript that addresses the points raised during the review process.

Authors should pay attention to minor revisions indicated by the reviewers.

We would appreciate receiving your revised manuscript by Apr 24 2020 11:59PM. To enhance the reproducibility of your results, we recommend that if applicable you deposit your laboratory protocols in protocols.io, where a protocol can be assigned its own identifier (DOI) such that it can be cited independently in the future. For instructions see: http://journals.plos.org/plosone/s/submission-guidelines#loc-laboratory-protocols

We look forward to receiving your revised manuscript.

Kind regards,

Jose M. Moran

Academic Editor

PLOS ONE

Reviewers' comments:

Reviewer's Responses to Questions

**Comments to the Author**

1. If the authors have adequately addressed your comments raised in a previous round of review and you feel that this manuscript is now acceptable for publication, you may indicate that here to bypass the “Comments to the Author” section, enter your conflict of interest statement in the “Confidential to Editor” section, and submit your "Accept" recommendation.

Reviewer #1: (No Response)

Reviewer #2: All comments have been addressed

2. Is the manuscript technically sound, and do the data support the conclusions?

Reviewer #1: Partly

Reviewer #2: Yes

3. Has the statistical analysis been performed appropriately and rigorously? 

Reviewer #1: No

Reviewer #2: Yes

4. Have the authors made all data underlying the findings in their manuscript fully available?

Reviewer #1: No

Reviewer #2: Yes

5. Is the manuscript presented in an intelligible fashion and written in standard English?

Reviewer #1: Yes

Reviewer #2: Yes

6. Review Comments to the Author

Reviewer #1: The new table 2 used a log2 tranformation of metal concentration, but the first column used uints without transformed, such as Pb(ug/dL), Al(ug/L), Cd(ug/L), Se(ug/dL), Zn(mg/L), were incorrected. That is why the negative numbers appears. Think about that, -2.15 ug/L As in blood, is so weird. If the authors used log2 transformation, then the units (ug/dL, ug/L, mg/L) in the table 2 should be ommitted. Otherwise, please used geometric means.

Reviewer #2: The authors addressed the comments and the manuscript is improved.

Still there are two suggestions:

1- Heavy metals. I previously commented that the trace elements measured by this research are not all "heavy metals", but there are semi-metallic elements and non-metallic elements. It is unclear why, following this comment, the authors transformed all the noun "metals" in the manuscript as "heavy metals". Please consider revising this.

2- Table 2 and log transformed data. Log transformation is a very common way to obtain normal distribution of data, useful for applying parametric statistics. This was properly done and I appreciate it. However, when reporting data in tables they should be reported as normal values. To report data in the log transformed form is not very clear for readers and should be avoided.

7. PLOS authors have the option to publish the peer review history of their article (what does this mean?). If published, this will include your full peer review and any attached files.

Reviewer #1: No

Reviewer #2: No

---

## [Author Response · Author response to Decision Letter 2]

24 Apr 2020

April 23, 2020

Dear Editor Jose Moran and Reviewers, 

I hope you are all well during this difficult time. I appreciate your continued work to improve our manuscript and feel that it has benefitted significantly from your contributions. I have thought carefully about your comments and responded as best I could. I hope you find the updated table and nomenclature acceptable for publication. As with the previous revision, I have included a document with all of the de-identified data to make the data underlying the published findings publicly available. I eagerly await to hear of next steps. 

Sincerely, 

Emily Moody

 

COMMENTS:

Reviewer #1: The new table 2 used a log2 tranformation of metal concentration, but the first column used uints without transformed, such as Pb(ug/dL), Al(ug/L), Cd(ug/L), Se(ug/dL), Zn(mg/L), were incorrected. That is why the negative numbers appears. Think about that, -2.15 ug/L As in blood, is so weird. If the authors used log2 transformation, then the units (ug/dL, ug/L, mg/L) in the table 2 should be ommitted. Otherwise, please used geometric means.

Thank you for this comment. I agree that units should have been omitted from that table and that using transformed data in this table is confusing. The table now includes the median and IQR for the entire cohort, and stunted and non-stunted subgroups and a Wilcoxan Rank Sum test for the test for differences in distribution. The mean and sd are included in the supplemental table for anyone interested in this information. 

Reviewer #2: The authors addressed the comments and the manuscript is improved.

Still there are two suggestions:

1- Heavy metals. I previously commented that the trace elements measured by this research are not all "heavy metals", but there are semi-metallic elements and non-metallic elements. It is unclear why, following this comment, the authors transformed all the noun "metals" in the manuscript as "heavy metals". Please consider revising this.

I appreciate this comment and have thought about this a lot. I struggled to find a noun that can simply and accurately describe the mixtures. The common usage in environmental chemistry and environmental health includes As and Se as “heavy metals” although this may be an antiquated and ultimately inaccurate term. I have removed the use of “heavy metal” and have continued to generally refer to the mixture as metals with the following explanation. Where possible, I have used other descriptors. 

“For simplicity, we refer to our mixture including lead (Pb), arsenic (As), cadmium (Cd), selenium (Se), and zinc (Zn), as metals, although Zn is a semi-metallic element and Se is a non-metallic element.”

2- Table 2 and log transformed data. Log transformation is a very common way to obtain normal distribution of data, useful for applying parametric statistics. This was properly done and I appreciate it. However, when reporting data in tables they should be reported as normal values. To report data in the log transformed form is not very clear for readers and should be avoided.

As in the comment above, I agree that this was not the correct choice. I have changed the table so that it presents the median and IQR for the entire cohort, and stunted and non-stunted sub groups and a Wilcoxan Rank Sum test for the test for differences in distribution. The mean and sd are included in the supplemental table.

---

## [Editor Report · Decision Letter 3]

29 Apr 2020

Environmental exposure to metal mixtures and linear growth in healthy Ugandan children

PONE-D-19-30717R3

Dear Dr. Moody,

We are pleased to inform you that your manuscript has been judged scientifically suitable for publication and will be formally accepted for publication once it complies with all outstanding technical requirements.

With kind regards,

Jose M. Moran

Academic Editor

PLOS ONE
---

## [Editor Report · Acceptance letter]

5 May 2020

PONE-D-19-30717R3 

Environmental exposure to metal mixtures and linear growth in healthy Ugandan children 

Dear Dr. Moody:

I am pleased to inform you that your manuscript has been deemed suitable for publication in PLOS ONE. Congratulations! Your manuscript is now with our production department. 

With kind regards,

on behalf of

Dr. Jose M. Moran 

Academic Editor

PLOS ONE